# Relationships between Change of Direction, Sprint, Jump, and Squat Power Performance

**DOI:** 10.3390/sports8030038

**Published:** 2020-03-19

**Authors:** Luis Suarez-Arrones, Oliver Gonzalo-Skok, Irene Carrasquilla, Jose Asián-Clemente, Alfredo Santalla, Pilar Lara-Lopez, F. Javier Núñez

**Affiliations:** 1Performance Department, FC Basel 1893, 4052 Basel, Switzerland; ljsuamor@upo.es (L.S.-A.); pilarlaralopez74@gmail.com (P.L.-L.); 2Physical Performance and Sport Research, Pablo de Olavide University, 41013 Sevilla, Spain; icargar@outlook.com (I.C.); josasicle@gmail.com (J.A.-C.); asanher@upo.es (A.S.); 3Faculty of Health Sciences, Univertity of San Jorge, 50830 Zaragoza, Spain; oligons@hotmail.com

**Keywords:** COD, velocity, CMJ, Abalakov, flywheel inertial device

## Abstract

The aim of the study was to investigate the relationships between countermovement jump (CMJ) height and inertial power in squat and sprint variables with change of direction (COD) performance. Fifty young healthy active males participated in the study. To determine these relationships, we carried out a 10-m linear sprint test (T 10 m), vertical jump tests (CMJ and CMJ Abalakov), an assessment of power relative to bodyweight in a flywheel squat (P_bw_), and 10-m COD sprints with two different turn types (COD-90° and COD-180°). T10 m showed statistically large and moderate correlations with T10 m COD-180° (r = 0.55) and T10-m COD-90° (r = 0.41), respectively. Moderate to large correlations between jumping height, linear sprinting, and sprints with COD were found (r = −0.43 to r = −0.59), and there were unclear correlations between jumping height and the loss of speed caused by executing COD (DEC-COD). P_bw_ showed a large correlation with CMJ Abalakov and CMJ jump height (r = 0.65 and r = 0.57, respectively), and a moderate and large correlation with T 10 m, T 10 m COD-180°, and T10 m COD-90° (r = −0.33, r = −0.38, and r = −0.54, respectively). Despite the existence of substantial correlations between variables, straight linear sprinting, jumping performance, CODs and squat power were, for the most part, separate motor qualities (R^2^ from 14% to 34%), suggesting that all of them should be specifically assessed and trained.

## 1. Introduction

Change of direction (COD) speed is a component of agility that describes movement wherein no reaction to a stimulus is required and the change of direction (COD) is previously planned [1,2]. This COD skill can be decisive for performance in many field and court sports [1,3,4]. Several CODs are executed during a team sports match, and players who can change direction fastest may have an advantage over their opponents during competition [3]. COD speed is dependent upon a number of factors, and previous studies suggest that straight sprinting speed and leg neuromuscular qualities could be important determinants in COD performance [1,5,6]. During a COD, it is necessary to execute force rapidly to accelerate, and to develop eccentric and concentric strength for decelerating and re-accelerating in the new direction. For this reason, it seems reasonable to think that there may be a relationship between strength/power, straight line sprint, and COD performance. 

Several studies have investigated the relationship between straight-line sprinting speed and COD tests [1,3,5,7,8,9], showing conflicting and inconsistent findings. In this regard, previous studies reflected large and very large relationships between COD performance (505 test, pro agility, Illionois) and acceleration ability and/or maximal sprinting speed [4,10]. Similarly, Buchheit et al. [11] found large to very large correlations between a 30-m linear sprint and a sprint with COD in different angles (45°, 90°, and 135°), and a more recent study showed very large correlations between 15-m linear sprinting and a sprint with two 60° CODs [12]. In contrast, Little and Williams [5], working with soccer players, found poor correlations between acceleration, maximum speed, and COD. Therefore, it was suggested that COD and linear sprint should be trained independently due to the low to moderate relationships between these abilities [13]. This finding was also supported in a previous study suggesting that for the most part (R^2^ < 50%), both are separate motor qualities [2]. 

Based on the relationships between linear sprint and strength, and also considering the forces generated during the contact phase in a COD, it was proposed that leg muscle qualities such as strength, power, and reactive strength could be important determinants of COD ability [1,2,3]. For this reason, the association between countermovement jump (CMJ) and drop jump (DJ) and COD was extensively studied without producing a clear consensus [6], while relationships with strength or power assessed through closed-kinetic chain exercises have been less examined [2]. While a group of studies show statistical relationships between COD and CMJ [4,6,10,14] and DJ performance [1,6,15], in contrast, others studies have shown lower relationships with no statistical associations with CMJ [8,13] and DJ [4,13,16]. In addition to this, it seems obvious that relative strength/power may be essential for COD performance [17]. Related to this, previous studies reflected statistical relationships between COD and 1RM relative to bodyweight during squat or leg press exercises in college and female athletes [8,17,18], although these relationships were weaker and unclear in male athletes [8,16]. Therefore, this information should be interpreted with caution. 

During COD actions, a series of stretch-shortening cycles are executed [6], and flywheel devices are a very convenient technology for emphasizing more forceful actions in the eccentric–concentric transition phase [19,20]. However, to our knowledge, there are no studies examining the relationships between the power during the squat exercise using inertial devices, jumping ability, acceleration during a linear sprint, and COD performance. Therefore, the aim of the present study was to investigate the relationships between CMJ height, inertial power in squat and sprint variables with COD performance.

## 2. Material and Methods

### 2.1. Experimental Design

To determine the relationships between acceleration, lower-limb explosive strength capacity, and COD performance, we carried out a 10-m linear sprint test, a vertical jump assessment (CMJ and countermovement jump Abalakov), an evaluation of the lower-limb power during the squat exercise, and a 10-m COD sprint with two different turn types (90°–180°). Physically active men were tested on two different days. Jump tests, linear sprints, and COD sprints were performed during the first day, and at least 72 h later the squat power was assessed.

### 2.2. Participants

Fifty young, healthy team-sport athletes (age: 22.7 ± 2.8 years; body mass: 75.4 ± 8.4 kg; height: 175.4 ± 8.3 cm) volunteered to participate in the present study. All subjects were actively training and practicing soccer, basketball, handball, and rugby 3–4 times per week on average. This study was approved by the Institutional Ethics Review Committee (Pablo de Olavide University, Seville, Spain). All participants were fully informed about the protocol and were required to give written consent in accordance Declaration of Helsinki II.

### 2.3. Testing Procedures

Prior to starting the experimental period, all participants were familiarized with the testing procedures to avoid any learning effects. All tests were performed indoors on a futsal court. During the first testing session, the athletes completed the tests in the following order: 1) jumping tests, 2) linear sprinting test, and 3) COD tests (90°–180°). Athletes performed a 7 to 8-min standardized warm-up protocol, which included jogging, joint mobility exercises, CODs, landings and 2 sets of 10 reps in the squat exercise. Additionally, prior to the beginning of each assessment, a specific warm-up for each test was performed, comprising 1 set of 5 reps (jumping tests) or 2 sub-maximal efforts (linear or COD sprints). In the second testing session (between 75–80 h later), lower-limb power during the squat exercise was assessed using an inertial device. After a similar standardized warm-up, including two sub-maximal sets, athletes performed 2 sets of 8 reps in the squat exercise. There was a 2-min rest between each set during which the subjects rested in a standing position. Care was taken to allow sufficient rest between all tests to limit the effects of fatigue on subsequent trials and tests. Subjects were asked not to perform any strenuous exercise during the day before each test, and they were also asked to follow a similar diet on the days of the test.

#### 2.3.1. Ten-Meter Linear Sprint Test

Athletes were assessed over a 10-m linear sprint test and time was recorded using photoelectric cells (Racetime2, Microgate^®^, Bolzano, Italy). The front foot was placed 0.5 m before the first timing gate, and players started when ready, thus eliminating reaction time. Subjects were given 2 practice trials performed at submaximal intensity after a thorough warm-up to familiarize them with the test. Three minutes after warm-up, two trials were completed, with two minutes of passive rest between trials, and the best performance 10-m trial was used for the subsequent statistical analysis (T10). 2.3.2. COD tests.

Two left and right 10-m COD sprints of 90° (COD-90°) and 180° (COD-180°) were carried out in left and right turns. COD-90° consisted of a 5-m sprint in a straight line and a right- or left-turn of 90° between 4 sticks (height: 1.5 m) placed vertically, with the aim of getting to the finish line as fast as possible (5 + 5-m). COD-180° consisted of a 5-m sprint in a straight line, touching a line with a foot (right or left), and a turn of 180° to come back to the starting line as fast as possible (5 + 5-m). The time was recorded with the same photoelectric cells (Racetime^2^, Microgate^®^, Bolzano, Italy)*,* with two photoelectric cells placed on the start and finish line during the COD-90°, and one photoelectric cell on the start/finish line for COD-180°. The front foot was placed 0.5 m before the first timing gate, and athletes started when ready, thus eliminating reaction time. Subjects executed two trials of each COD sprint test and the fastest was retained for calculations (T 10 m COD). The mean time between the best result for the right and for the left in both COD tests (i.e., COD-90° and COD-180°) was compared to the fastest 10-m straight-line sprint time. The loss of speed caused by executing COD (DEC-COD) was calculated as a percentage using the formula: [(T 10 m COD − T 10 m)/T 10 m) × 100]. 

#### 2.3.2. Vertical Jump Tests

CMJ and CMJ-Abalakov [21] tests were used to maximize stretch-shortening cycle activity and to assess the explosive strength of the lower extremity muscles. Both tests were performed using an infrared contact platform (Optojump^®^, Microgate, Bolzano, Italy). During the CMJ, subjects were instructed to keep their hands on their hips, with the depth of the countermovement self-selected [22], while during the CMJ-Abalakov test, arm swim was allowed during the jump [21]. All subjects were instructed to land in an upright position and to bend the knees after landing. Each test was performed three times, separated by 45 s of passive recovery, and the best jump performed was used for the subsequent statistical analysis for each type of vertical jump. 

#### 2.3.3. Lower Limb Power Test 

Lower limb power was assessed using the squat exercise with a non-gravity-dependent flywheel inertial device (Exxentric kBox, Exxentrix AB, Stockholm, Sweden), allowing subjects to perform maximal concentric (CON) and eccentric (ECC) actions [23]. Two sets of all-out 8 repetitions (0.10 kg/m^2^ moment inertia), with a 2-min rest between each set, were monitored. The best mean power relative to bodyweight (P_bw_) was considered for subsequent analysis. Each set start with two sub-maximal actions followed by six maximal repetitions. Subjects were requested to push maximally through the entire range of motion of the CON action. During the subsequent ECC phase, subjects were requested to gently resist during the initial part (first 20°–30°) and to resist maximally, aiming to bring the wheels to a stop, at a ~110° knee angle before initiating the next cycle. As in previous studies with these inertial devices [24,25], power and velocity during the testing session were sampled at 100 Hz using a rotatory encoder (SmartCoach^TM^, SmartCoach Europe AB, Stockholm, Sweden) and associated software (SmartCoach^®^ v.5.2.0.5). 

### 2.4. Statistical Analyses

Data are presented as mean ± standard deviation (SD). The normality of distribution of the variables in the Pre-test and the homogeneity of variance across groups were verified using the Shapiro–Wilk test and Levene’s test, respectively. A reliability analysis (intraclass correlation coefficient; ICC) was carried out for all within-session outcome measures. Pearson’s correlation coefficients were calculated to establish the respective relationships between jumping tests, linear sprinting test, flywheel squat power test and COD tests. The magnitude of the correlation (r) between test measures was assessed with the following thresholds: ≤0.1, trivial; >0.1–0.3, small; >0.3–0.5, moderate; >0.5–0.7, large; >0.7–0.9, very large; and >0.9–1.0, almost perfect [26]. If the 90% confidence interval overlapped small positive and negative values, the magnitude of the correlation was deemed unclear; otherwise, the magnitude was deemed to be the observed magnitude [26]. We also postulated that if jumping ability, linear sprinting ability, flywheel squat power, and CODs are dependent and exist as general qualities, rather than a specific quality, individuals would rank similarly despite the different tests. The appropriate statistical test to validate the concept of generality has been suggested to be a correlation coefficient of r = 0.71 or greater [27], as this degree of association would suggest a minimum of 50% common variance (R^2^; coefficient of determination) [11]. For further analysis, players were divided into 2 groups, fastest and slowest, based on a moderate standardized difference (ES) (i.e., 0.2 × between-groups SD) from the group average as in previous studies [28,29]. The standardized difference or effect size (ES, 90% confidence limit (90% CL)) between the fastest and slowest players in the loss of speed due to executing a COD was calculated. The threshold values for assessing the magnitudes of the ES (changes as a fraction or multiple of baseline standard deviation) were <0.20, 0.20, 0.60, 1.2, and 2.0 for trivial, small, moderate, large, and very large, respectively [26]. The chances of greater or smaller differences than the smallest worthwhile difference were assessed qualitatively as follows: 25% to 75%, possible; 75% to 95%, likely; 95% to 99%, very likely; and >99%, almost certain [26]. 

## 3. Results

The assessments of the best attempts in vertical jumps, linear sprinting, sprints with COD, and power in the flywheel squat exercise, together with their reliability, are shown in Table 1. 

Linear regression analyses between the linear sprint and sprint with COD are shown in Figure 1. T 10 m showed large and moderate statistical correlation with T10 m COD-180° (r = 0.55, p < 0.05) and T 10 m COD-90° (r = 0.41, p < 0.05), respectively (Figure 1A). T10 was moderately statistically correlated with DEC-COD 180° and DEC-COD 90° (r = −0.48 and r = −0.45, p < 0.05, respectively).

Linear regression analysis between jumping height, linear sprinting, and sprints with COD are shown in Figure 2. Moderate to large statistical correlations between jumping height, linear sprinting, and sprints with COD were found (r = −0.43 to r = −0.59, p < 0.05) (Figure 2A–C, respectively). There were unclear correlations between jumping height and DEC-COD (p > 0.05). 

Linear regression analyses between COD and P_bw_ in squat, jumping height, linear sprint, and sprints are shown in Figure 3. P_bw_ showed a large statistical correlation with CMJ-Abalakov and CMJ height (r = 0.65 and r = 0.57, p < 0.05, respectively) (Figure 3A). P_bw_ showed moderate and large statistical correlations with T 10 m, T10 m COD-180°, and T10 m COD-90° (r = −0.33, r = −0.38, and r = −0.54, p < 0.05, respectively). There were unclear correlations between P_bw_ and DEC-COD (p > 0.05).

The fastest athletes lost more time executing the COD than their slower counterparts in COD-180° (−47.1 ± 5.7% vs −41.1 ± 5.6%, ES = −1.06 ± 0.47 (almost certainly), respectively) and COD-90° (−37.0 ± 7.9% vs −31.3 ± 5.4%, ES = −0.84 ± 0.46 (very likely), respectively). 

## 4. Discussion

The aim of the study was to investigate the relationships between CMJ height, inertial power in squat and sprint variables with COD performance. The main finding of our results, despite the existence of statistical correlations, was that straight linear sprinting, jumping performance, CODs and squat power were, for the most part, separate motor qualities (R^2^ from 14% to 34%). 

The relationships found in the present investigation revealed moderate to large statistical correlations between linear straight sprinting and COD tests (COD-90° and COD-180°). Due to the lack of a “gold standard” COD test, there is a wide variety of COD tests used in the current scientific literature; therefore, it is difficult to make direct comparisons with the present results. However, the associations shown in the present study are within the range (r = 0.22 to 0.70) of previously published reports in male team-sports (professional and college) players [5,11,13,30,31,32]. While there are only two studies that have reported higher levels of association between short linear sprints (10 to 20 m) and COD tests [11,30], our results show greater correlations than the vast majority of team-sports studies focusing on this topic [5,13,31,32]. It should be noted that those large correlations were found in COD tests where the total distance covered during the test was greater than 20 m (30 m to 40 m). Furthermore, the differences in the number of CODs, COD angles, and the distances covered prior to COD could be the main reason for the apparent between-study differences. Despite these between-test differences, no study has achieved the generality criterion (r = 0.71; R^2^ < 50%) to consider linear sprinting and CODs as a general quality in senior male team-sport athletes [2,11,27]. Based on these findings, linear sprinting and CODs seem to be separate motor qualities in these athletes. The variations in sprint distance, the number of turns and turning angles associated with each COD test, adjustments in stride pattern to decelerate and reaccelerate, force application in different axes, and the biomechanical and neuromuscular differences compared to linear sprinting [2,11] may explain a great deal of the differences. Consequently, these abilities should be trained and assessed independently.

Unlike other studies, the present investigation analyzed the DEC-COD in order to establish whether the fastest athletes lost more time executing the COD than their slower counterparts. The correlational analyses showed a statistical relationship (r = −0.45 to −0.48) between T 10 m and DEC-COD (90° and 180°), and the results of the present study show that the fastest athletes demonstrated a greater DEC-COD than their slower counterparts in both COD-180° (–47.1% vs –41.1%, respectively) and COD-90° (–37.0% vs –31.3%, respectively). From a practical point of view, in those cases where players showed a high DEC-COD, complementary training in order to improve COD should be prescribed by coaches. 

In the present study, we observed that bilateral jumping performance (i.e., CMJ and CMJ-AbkJ) had a moderate to large statistical correlation with COD-180° and COD-90° tests. These results are in accordance with those reported in male team-sports players (e.g., college or physical education students, r= −0.39 to −0.59) in tests that involved similar COD angles to our study (90° or 180°) and only demanded a single COD [4,6,33]. However, other studies have provided poorer correlations (r= −0.15 to −0.38) [8,13,31]. Differences in the number of CODs (1 vs. 3–4), COD angles (90°–180° vs. 45°–135°), running mechanics (forward vs. lateral-shuffles) or COD technique (sharp vs. rounded) may be the reason for these between-study differences. The number of CODs may influence the magnitude of the association, and, hence, studies involving a single COD are not comparable to those with a higher number of CODs where the learning process and technique may have more impact. Furthermore, the subjects involved in each study (professional or highly trained team-sports players vs. college or physical education team-sports players) could be another differentiating factor. However, despite achieving close relationships when COD tests were similar to ours and as occurred in the linear straight sprinting, no study with senior male team-sports players has reached the cut-off required to establish a general ability (r = 0.71; R^2^ < 50%) [2,11,27]. It is likely that those actions that are unilaterally performed may demand similar requirements to the actions involved in every COD and, in consequence, it is possible that unilateral actions such as unilateral jumps or unilateral power (i.e., horizontal, lateral, or vertical) may have achieved greater correlations. While it seems that COD improvement in specific conditions (angles and number of CODs) may be helped by better bilateral jumping performance [2], unilateral jumping performance could be a better predictor. However, this is only a hypothesis, and our findings do not support this assumption. Thus, based on the present results and those of a previous study [6], both abilities should be analyzed individually. 

This is the first study that has analyzed the statistical relationships between the power output in a flywheel inertial device and COD; therefore, direct comparisons with other studies are not possible. However, it has been reported that these devices are designed to emphasize more forceful actions in the ECC-CON transition phase [19,20], being able to produce a better activation during two different COD tasks (side-step and crossover cutting) [34]. Furthermore, COD was improved in soccer players after 11 weeks of flywheel training with these devices [35]. Based on these findings, the power output obtained in these flywheel devices may be related to COD ability, but the statistical relationships found in the present study were moderate to large (r = −0.38 to −0.54), without achieving the generality criterion. These findings are in line with those of other strength assessments (1RM relative to body mass) (r = −0.33 to −0.45) [4,8], even though studies that have used other measurements, such as isokinetic eccentric strength, have reported greater associations (r = −0.63 to −0.78) in male senior team-sports players [4,30]. In the latter case, these differences may be due to the complexity of the COD test used (single vs. several CODs) or the velocity employed in the isokinetic testing (60°/s vs.180°/s). Indeed, as the number of CODs increases, the influence of eccentric strength measured at a similar velocity to the COD task may be of greater magnitude (players accelerate, brake, and re-accelerate). It is also possible that unilateral repeated ECC-CON assessment at different loads could be recommended as a better alternative and, consequently, as a better predictor. Further studies are needed to investigate this hypothesis.

## 5. Conclusions and Practical Applications

Our data reveal that the abilities tested (i.e., jumping, linear sprinting, COD, squat power) are separate motor qualities, suggesting that all of them should be specifically assessed and trained. In addition, a new variable (DEC-COD) should be included in team-sport testing in order to detect players’ deficits in this capacity. Providing this information to strength and conditioning coaches will enable the development of training prescriptions and practical recommendations to improve players’ locomotor profiles.

## Figures and Tables

**Figure 1 sports-08-00038-f001:**
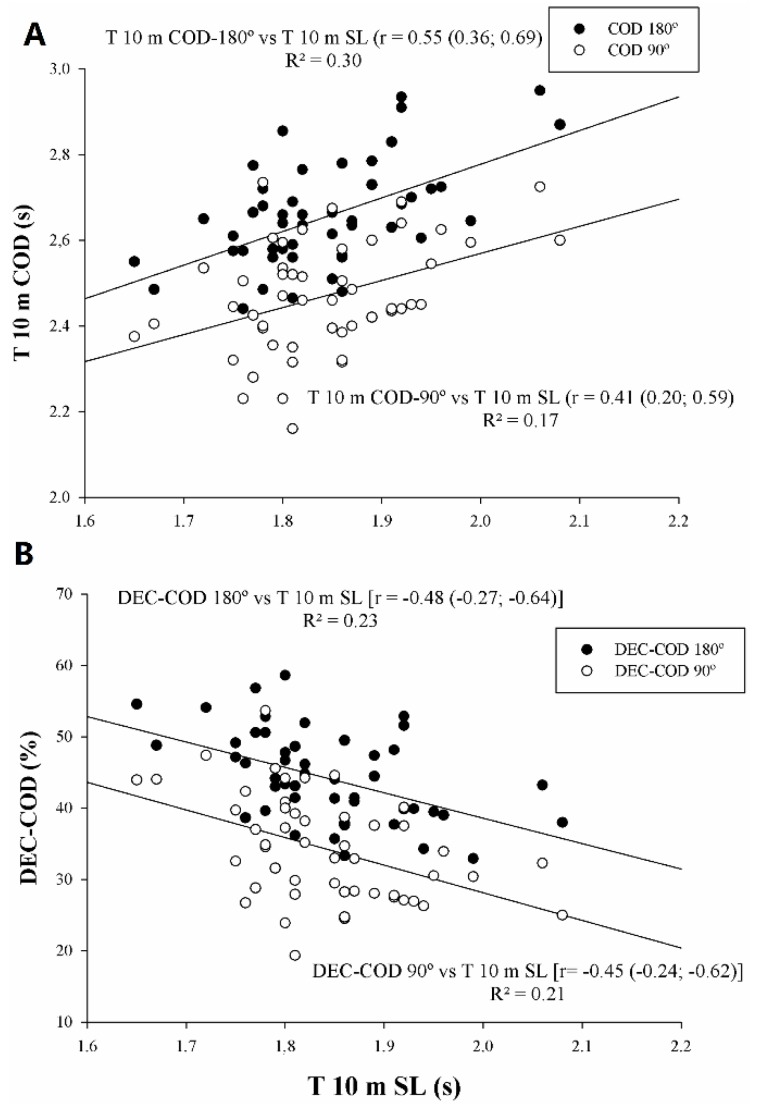
Relationships between linear sprinting (T 10 m) and sprints with change of direction (COD) (90% CL). DEC-COD: percentages mean speed loss due to execute COD. (**A**) T 10 m with T 10 m COD 180° and T 10 m COD 90°; (**B**) T 10 m with DEC-COD 180° and DEC-COD 90°.

**Figure 2 sports-08-00038-f002:**
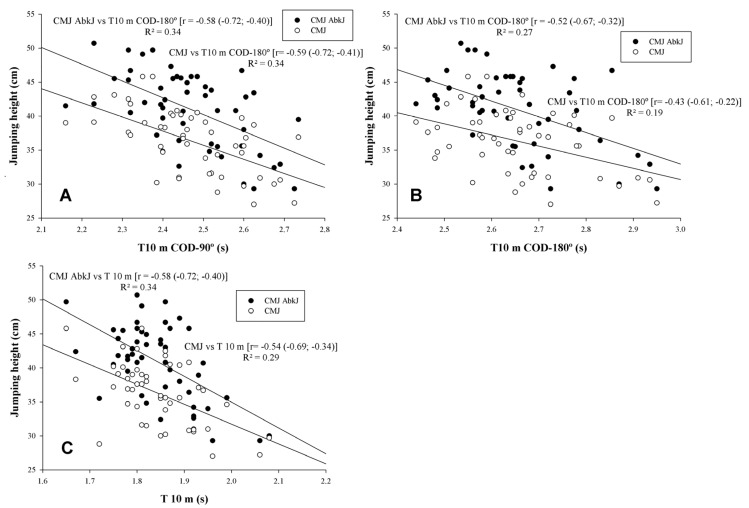
Relationships between jumping height, linear sprinting (T 10 m), and sprints with change of direction (COD) (90% CL). CMJ: Countermovement jump; CMJ-AbkJ: Abalakov jump. (**A**) CMJ and CMJ-AbkJ with T 10 m COD 90°; (**B**) CMJ and CMJ-AbkJ with T 10 m COD 180°; (**C**) CMJ and CMJ-AbkJ with T 10 m.

**Figure 3 sports-08-00038-f003:**
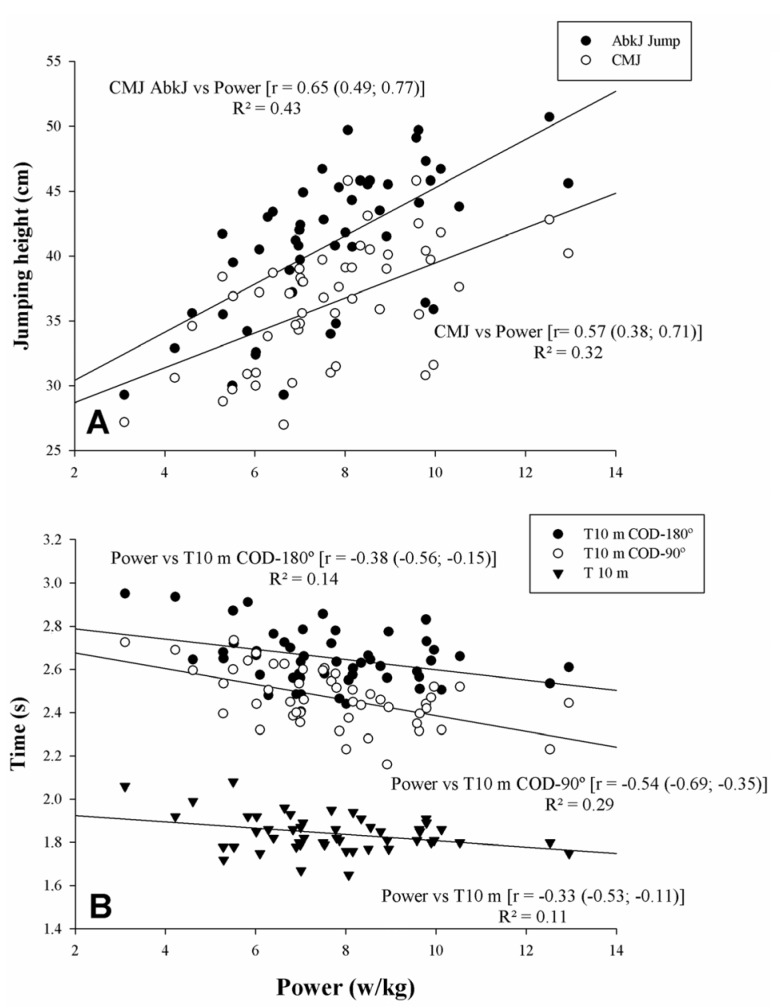
Relationships between relative power to bodyweight in squat, jumping height, linear sprint, and sprints with change of direction (COD) (90% CL). CMJ: Countermovement jump; CMJ-AbkJ: Abalakov jump. (**A**) CMJ and CMJ-AbkJ with Power; (**B**) T 10 m with T 10 m COD 180° and T 10 m COD 90°.

**Table 1 sports-08-00038-t001:** Measures of reliability for jumps, linear sprinting, sprint with change of direction (COD), and flywheel squat power.

	CMJ	CMJ-AbkJ	T 10 m SL	T 10 m COD-180°-R	T 10 m COD-180°-L	T 10 m COD-90°-R	T 10 m COD-90-L	P_bw_
Mean ± SD	36.4 ± 4.6 cm	41.0 ± 5.6 cm	1.84 ± 0.10 s	2.66 ± 0.13 s	2.64 ± 0.13 s	2.47 ± 0.14 s	2.47 ± 0.14 s	7.69 ± 1.95 w/kg
TE (90% CL)	0.53 (cm)	0.65 (cm)	0.03 (s)	0.04 (s)	0.03 (s)	0.03 (s)	0.03 (s)	0.24 (w/kg)
CV (90% CL)	1.6 (%)	1.7 (%)	1.5 (%)	1.4 (%)	1.1 (%)	1.4 (%)	1.0 (%)	2.5 (%)
ICC (90% CL)	0.98 (0.99; 0.99)	0.99 (0.99; 0.99)	0.89 (0.79; 0.94)	0.92 (0.86; 0.96)	0.96 (0.92; 0.98)	0.94 (0.88; 0.97)	0.97 (0.95; 0.99)	0.99 (0.98; 0.99)

CMJ: Countermovement jump; CMJ-AbkJ: CMJ Abalakov jump; T 10 m SL: 10 m time (straight-line sprint); T 10 m COD: 10 m time with change of direction; R: right; L: left; P_bw_: Relative power to bodyweight. TE: Typical error of measurement; CV: TE expressed as a coefficient of variation; ICC: Intraclass correlation coefficients.

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
