# Peer review of "Relationships between Change of Direction, Sprint, Jump, and Squat Power Performance"

_sports, 2020, doi:10.3390/sports8030038_

Round 1

Reviewer 1 Report

[Sports] Manuscript ID: sports-732592

This is a well written and well-presented study (although the quality of the figure prints in the manuscript, unfortunately, was poor, which made them hard to read). The study investigates associations between different power-related variables that are often used to test athletes. This has been done in many studies before and in this respect the study is not very new. Indeed, to a great extent, the authors find associations that are roughly in line with those reported in other studies. They also provide reasonable explanations for the range in associations found in the literature (for instance the number of COD movements used varies among different studies). The new aspects of the present study seem to be the inclusion concentric eccentric power assessed with a flywheel squatting device and the presentation of a new (please explicitly state in the manuscript to what extent this variable is truly new) variable: DEC-COD, which indicates the percentage of speed loss when 10m distance has to be covered with a COD compared to without COD. I can see that this can be an important measure in practice to assess the COD-capacity of athletes.

My main concern is that the authors repetitively state  that they aim to investigate to what extent the different variables can be used as predictors of COD performance. As I see it they merely investigate associations, there are no predictions made (e.g. the regression models are not used to make predictions in a different set of subjects). Is it necessary to use the term predictors? Reporting the correlations (and their squared values as indicators for the variance explained) seems enough. The aim could be rephrased to something like: to investigate the relations between CMJ jump height, squat power and sprint variables with COD performance. There also are no indications that the authors tried to combine different variables (e.g. 10 m sprint AND CMJ) into one regression equation to investigate whether this would increase the proportion of explained variance in COD. Was this attempted or deemed not be important?

Other remarks.

Abstract: DEC-COD has not been introduced, only after reading the entire manuscript I understood what was meant by this acronym. Please correct.

Introduction line 39: Why would one want to predict COD performance from linear sprint AND/OR leg strength or power? Especially from a practical point of view, the straightforward thing to do would be to measure COD performance directly, or wouldn’t it?

Line 81 please state that the participants were male

Line 95 what is COS? And what are landings?

What is the added value of investigating both CMJ and CMJ-Abalakov? (By the way, I associate Abalakov with a jump and reach test…..why not use CMJ with and without arm swing instead?)

I expect a very high correlation between both tests (not reported, please add) and therefore I do not see the added value of investigation both associations with COD. (In that respect it would have been more useful to investigate CMJ and squat jump)

Results

Table 1 . The reliability measures should be explained in the method section. Which ICC model was used? etc. It is important to state that these measures were all within-session measures, I assume they are? (which generally will be higher than between session measures)

Discussion line 80 (the flywheel) It seems that task execution of the squat with the flywheel seems rather complicated: ‘gently resist during the first part ….etc.’ Doesn’t this require more practice and to what extent may this have affected the findings? Are the power values reported in line with earlier reports?

The performance range among the participants seems quite high. This will increase the correlations (and ICC) measures. Please comment and it may be useful to provide some additional participant information in the methods which would clarify the rather broad performance range. Were there large differences in BMI, training status? etc.

Author Response

Sports BASEL. Round 1.

Reviewer 1

This is a well written and well-presented study (although the quality of the figure prints in the manuscript, unfortunately, was poor, which made them hard to read). The study investigates associations between different power-related variables that are often used to test athletes. This has been done in many studies before and in this respect the study is not very new. Indeed, to a great extent, the authors find associations that are roughly in line with those reported in other studies. They also provide reasonable explanations for the range in associations found in the literature (for instance the number of COD movements used varies among different studies). The new aspects of the present study seem to be the inclusion concentric eccentric power assessed with a flywheel squatting device and the presentation of a new (please explicitly state in the manuscript to what extent this variable is truly new) variable: DEC-COD, which indicates the percentage of speed loss when 10m distance has to be covered with a COD compared to without COD. I can see that this can be an important measure in practice to assess the COD-capacity of athletes.

My main concern is that the authors repetitively state  that they aim to investigate to what extent the different variables can be used as predictors of COD performance. As I see it they merely investigate associations, there are no predictions made (e.g. the regression models are not used to make predictions in a different set of subjects). Is it necessary to use the term predictors? Reporting the correlations (and their squared values as indicators for the variance explained) seems enough. The aim could be rephrased to something like: to investigate the relations between CMJ jump height, squat power and sprint variables with COD performance. There also are no indications that the authors tried to combine different variables (e.g. 10 m sprint AND CMJ) into one regression equation to investigate whether this would increase the proportion of explained variance in COD. Was this attempted or deemed not be important?

 Answer: Authors would like to thank reviewer his/her labor on the manuscript at this first stage of the review process. Thanks for your contribution and comments acknowledge. In fact that is what we are trying to do. We re-phrase the aim, so that it now reads:” the aim of the present study was to investigate the relationships between CMJ jump height, inertial power in squat and sprint variables with COD performance”. The relations between others variables have been done in several studies before, so we decide focus on the relations between the variables (including the same distance in linear sprint) with a simple left and right 10-m COD (90º & 180º).

Other remarks.

Abstract: DEC-COD has not been introduced, only after reading the entire manuscript I understood what was meant by this acronym. Please correct.

Answer: Thanks for your contribution. Changes have been included in the abstract.

Introduction line 39: Why would one want to predict COD performance from linear sprint AND/OR leg strength or power? Especially from a practical point of view, the straightforward thing to do would be to measure COD performance directly, or wouldn’t it?

Answer: Comment acknowledge. We agree with the proposal so this information has been removed.

Line 81 please state that the participants were male

Answer: Thanks for your contribution. We re-phrase it, so that it now reads:” Physical active men were testing in two different…”

Line 95 what is COS? And what are landings?

Answer: Thanks for your contribution. There was an error. We re-phrase COS by CODs. Landings are exercises in which the subjects trying to reduce the body impact after a jump or to decelerate lineal movement in different directions, favoring the neuro-muscular adaptation or preventing an injury during COD, sprint or jump performance.

What is the added value of investigating both CMJ and CMJ-Abalakov? (By the way, I associate Abalakov with a jump and reach test…..why not use CMJ with and without arm swing instead?)

Answer: Thank you for your consideration. Under our point of view Abalakov Jump is a CMJ with arms swing instead. In fact we describe in the article (Line 140) as: "During the CMJ, subjects were instructed to keep their hands on their hips, with the depth of the countermovement self-selected [22], while during the CMJ-Abalakov test, arm swim was allowed during the jump [21]".

I expect a very high correlation between both tests (not reported, please add) and therefore I do not see the added value of investigation both associations with COD. (In that respect it would have been more useful to investigate CMJ and squat jump)

Answer: Comment acknowledge. Related this question and in addition to the response given above, the relations between CMJ and Abalakov have been showed in previous studies (Rodriguez-Rosell et al, Traditional vs. Sport-Specific Vertical Jump Tests: Reliability, Validity, and Relationship With the Legs Strength and Sprint Performance in Adult and Teen Soccer and Basketball Players. J Strength Cond Res. 2017 Jan;31(1):196-206; Nikolaidis et al. Who jumps the highest? Anthropometric and physiological correlations of vertical jump in youth elite female volleyball players. J Sports Med Phys Fitness. 2017 Jun;57(6):802-810.), but we decide focus on others relationships. According with your consideration, in future studies we will introduce the squat jump variable.

Results

Table 1 . The reliability measures should be explained in the method section. Which ICC model was used? etc. It is important to state that these measures were all within-session measures, I assume they are? (which generally will be higher than between session measures)

Answer: Thanks for your contribution. There was an error. We introduce a phrase to explain the reliability measures, so that it now reads:A reliability analysis (intraclass correlation coefficient; ICC) was carried out for all within-session outcome measures”.

Discussion line 80 (the flywheel) It seems that task execution of the squat with the flywheel seems rather complicated: ‘gently resist during the first part ….etc.’ Doesn’t this require more practice and to what extent may this have affected the findings? Are the power values reported in line with earlier reports?

Answer: Thanks for your contribution. As we indicated in the methods section (Line 94): "Prior to starting the experimental period, all participants were familiarized with the testing procedures to avoid any learning effects".  The intraclass correlation coefficient (ICC) was no different with the others test proposed, and the coefficient of variation showed normal values for a squat test.

The performance range among the participants seems quite high. This will increase the correlations (and ICC) measures. Please comment and it may be useful to provide some additional participant information in the methods which would clarify the rather broad performance range. Were there large differences in BMI, training status? etc.

Answer: Thanks for your contribution. Logically the young healthy team-sport athletes which training 3–4 times per week, guarantees high performance. There were no differences in BMI or in training status.

Reviewer 2 Report

The article “Relationships between Change of Direction, Sprint, Jump, and Squat Power Performance” aimed to determine whether linear sprinting, jumping ability or 11  flywheel half-squat power can be predictors of change of direction (COD) performance. The authors have a simple study design, good results and some issues with the writing. Specially with the conclusions, which does not respond the aims of the article. Despite the issues, there are no fatal flaws and should be reconsidered after major review. See specific appoints below.

Major

  1. Introduction: despite the previous research, it is not clear why is so important to have a fast and low-cost field test to predict another fast and low-cost field test.
  2. L81: please provide more information about the participants. Such as competition level, what kind of sports they were athletes.
  3. L97: what was the maximum time between two sessions before it could be considered a dropout, or have a time effect on their results?
  4. L152: Since the authors have multiple variables to correlate with COD, a multiple regression using COD as dependent would be adequate. This method would test if is associated and could predict COD at once, instead of running multiple correlation tests.
  5. Table 1: author did not mention ICC in Statistical Analysis. Neither if is single or multiple measures.
  6. Figures: maybe it is generated PDFs, but the quality of figures is terrible. It is almost impossible to read.
  7. Discussion L25-28: the article’s objective is whether some tests could predict COD. Thus, in this paragraph you should address to this objective based on your results. Can any of the tests predict COD? You can explain your answer in the following paragraphs.
  8. Some rationale for clustering the athletes by their max speed should also be added in the intro section.
  9. The authors concluded that “the abilities tested (i.e., jumping, linear sprinting, COD, half-squat power) are separate motor qualities, suggesting that all of them should be specifically assessed and trained”, but this was not the objective of the article.
  10. I strongly suggest the authors to also include some “practical applications” paragraph before the conclusion. This is such a practical study for coaches and athletes and makes no sense to present the information and no saying to the public how they should use it.

Minor

  1. L30: which court sports?
  2. L52 to L72: recommend splitting this paragraph in two. Maybe in L65.
  3. 180: you mean “Linear regression”?
  4. Results L12: “Lineal regression”. Please double check this throughout the manuscript.

Author Response

Sports BASEL. Round 1.

Reviewer 2

Comments and Suggestions for Authors

The article “Relationships between Change of Direction, Sprint, Jump, and Squat Power Performance” aimed to determine whether linear sprinting, jumping ability or  flywheel half-squat power can be predictors of change of direction (COD) performance. The authors have a simple study design, good results and some issues with the writing. Specially with the conclusions, which does not respond the aims of the article. Despite the issues, there are no fatal flaws and should be reconsidered after major review. See specific appoints below.

Answer: Authors would like to thank reviewer his/her labor on the manuscript at this first stage of the review process. We have substantially amended the paper to clearly present the aims to the reader. We hope we have clearly stated our findings and conclusions.

Major

1. Introduction: despite the previous research, it is not clear why is so important to have a fast and low-cost field test to predict another fast and low-cost field test.

 Answer: Thanks for your contribution. The aim of the present study was to investigate the relations between CMJ jump height, inertial power in squat and  sprint variables with COD performance. One of the new aspects of the present study  was the inclusion concentric-eccentric power assessed with a flywheel device. These devices are usually employed during the strength training protocols in team sports athletes. Other aspect introduced in this study was the used of the percentage of speed loss when 10m distance has to be covered with a COD compared to without COD. From our point of view, the changes in this variable could be an additional parameter representative of the COD performance in order to detect players’ deficits in this independent capacity. The main finding of our results, despite the existence of statistical correlations, was that straight linear sprinting, jumping performance, CODS and half-squat power were separate motor qualities, suggesting that all of them should be specifically assessed and trained. In addition, a novel finding was that the fastest athletes lost more time executing the COD than their slower counterparts.

2.L81: please provide more information about the participants. Such as competition level, what kind of sports they were athletes.

 Answer: Thanks for your contribution. We re-phrase it, so that it now reads:”All subjects practice soccer, basketball, handball and rugby and were actively training 3–4 times per week on average”

3.L97: what was the maximum time between two sessions before it could be considered a dropout, or have a time effect on their results?

 Answer: Thanks for your contribution. All participants begin the second testing session between 75-80 hours later that finish the first testing session. We re-phrase it, so that it now reads:” In the second testing session (Between 75-80 hours later),…”  

4.L152: Since the authors have multiple variables to correlate with COD, a multiple regression using COD as dependent would be adequate. This method would test if is associated and could predict COD at once, instead of running multiple correlation tests.

Answer: Thanks for your contribution. There was an error. The aim of the present study was to investigate the relations between CMJ jump height, inertial power in squat and  sprint variables with COD performance.

5.Table 1: author did not mention ICC in Statistical Analysis. Neither if is single or multiple measures.

Answer: Comment acknowledge. We have introduced a sentence explaining the reliability measures, so that it now reads:A reliability analysis (intraclass correlation coefficient; ICC) was carried out for all within-session outcome measures”.

6.Figures: maybe it is generated PDFs, but the quality of figures is terrible. It is almost impossible to read.

Answer: Sorry, they were sent in TIFF format with good quality. We attach them again and also in jpeg.

7.Discussion L25-28: the article’s objective is whether some tests could predict COD. Thus, in this paragraph you should address to this objective based on your results. Can any of the tests predict COD? You can explain your answer in the following paragraphs.

Answer: Thanks for your contribution. In fact that is not what we are trying to do. We re-phrase the aim, so that it now reads:” the aim of the present study was to investigate the relations between CMJ jump height, inertial power in squat and  sprint variables with COD performance”.

8.Some rationale for clustering the athletes by their max speed should also be added in the intro section.

Comment acknowledge. Players were divided into 2 groups, fastest and slowest (based on a moderate standardized difference (ES) (ie, 0.2 × between-groups SD) from the group average as in previous studies) in order to know if the fastest athletes lost more time executing the COD than their slower counterparts. To our knowledge, this is a novel finding which no one has investigated yet due to employ more complex COD test.

9.The authors concluded that “the abilities tested (i.e., jumping, linear sprinting, COD, half-squat power) are separate motor qualities, suggesting that all of them should be specifically assessed and trained”, but this was not the objective of the article.

Answer: Thanks for your contribution. In fact that is what we are trying to do. We re-phrase the aim, so that it now reads:” the aim of the present study was to investigate the relations between CMJ jump height, inertial power in squat and sprint variables with COD performance”.

10.I strongly suggest the authors to also include some “practical applications” paragraph before the conclusion. This is such a practical study for coaches and athletes and makes no sense to present the information and no saying to the public how they should use it.

Answer: Comment acknowledge. With the results obtained in this study, our conclusions were the best practical applications that we could offer for coaches and athletes too. Changes have been made in the conclusions section (conclusion and practical applications). Our data revealed that the abilities tested (i.e., jumping, linear sprinting, COD, half-squat power) are separate motor qualities, suggesting that all of them should be specifically assessed and trained; and a new variable (DEC-COD) should be included in team-sport testing in order to detect players’ deficits in this independent capacity.

Minor

1.L30: which court sports?

Answer: Thanks for your contribution. Thanks for your contribution. We refer to indoor sport

2.L52 to L72: recommend splitting this paragraph in two. Maybe in L65.

Answer: Thanks for your contribution. Changes have been made.

3.180: you mean “Linear regression”?

Answer: yes, comment acknowledge and thanks for your contribution.

4.Results L12: “Lineal regression”. Please double check this throughout the manuscript.

Answer: comment acknowledge and thanks for your contribution.

Round 2

Reviewer 1 Report

No further comments.

Reviewer 2 Report

No further comments.